# OpenReview forum: "Measuring Bias of Web-filtered Text Datasets and Bias Propagation Through Training"
_ICLR.cc/2025/Conference — Submitted to ICLR 2025_

### Official Review · Reviewer_UcRC · 2024-10-26

**Soundness:** 3
**Presentation:** 2
**Contribution:** 3
**Rating:** 6
**Confidence:** 3

**Summary:**

This method proposes an interesting method to measure bias of web-filtered text datasets, and evaluate the bias propagation through training the large language models. The idea is insightful and the experiments are in general solid, but there are still several concerns to be addressed.

**Strengths:**

1. The idea is interesting and the problem of data bias in large language models pre-training dataset is an important challenge in the community
2. The proposed evaluation method is solid and reasonable/

**Weaknesses:**

1. The evaluation part needs to be more thorough, questions are listed in the next section
2. The architecture of the paper could be better organized for improving readability.

**Questions:**

1. Regarding the classification accuracy, it is unclear whether these authors choose these dataset combinations for classification experiments. I suggest the authors do a 2-way classification with respect to each dataset pairs, leading to a  matrix or heatmap showing the 2-way classification accuracies between all dataset pairs. This would give a clearer picture of which datasets are most/least distinguishable from each other.
2. The conclusion of dataset bias is valid, but could the authors do more investigation on the critical differences that differentiate between different datasets ? For example, changing some paraphrases in Category 1 may alter the classifier results to Category 2, thus these paraphrases may be a bias in Category 1. I could understand that it is hard to enumerate over all data samples, but some interpretable examples will be appreciated, such as a few concrete examples of text that are particularly indicative of each dataset.
3. The last section seems to be a draft without comprehensive evaluation. Some details are not clear, for example, the prompt for sentence generation from these LMs, different prompts on the impact of the classification accuracy. I would suggest a more structured evaluation framework for this section, such as a comparison of results across different models or datasets.

---

> ### Author Response · Authors · 2024-11-22
> **Response to reviewer UcRC**
>
> Many thanks for mentioning that our idea is interesting, that we address an important challenge in the community, and that our proposed evaluation method is solid and reasonable.
>
> &nbsp;
>
> Response to reviewer UcRC’s concerns:
>
> - Question 1: The reviewer suggests doing 2-way classification experiments on all possible combinations between the datasets. Thanks for the suggestion, we did those experiments and the results are in Appendix B of the revised paper.
>
> &nbsp;
>
> - Question 2: Regarding ``do more investigation on the critical differences that differentiate between different datasets’’, we conducted three more experiments to investigate what differentiates the datasets, the results are in Sections 4.4, 4.5, and 4.6 in the revised paper.
>
>     &nbsp;
>
>     In section 4.6 and appendix D in the revised paper, we also explain and show some particularly distinct examples that are unique to DCLM and FineWeb-Edu. For other datasets like C4 and FineWeb,  the distinguishing features are not as obvious, and require careful observation of many examples to notice the subtle differences in content and format.
>
> &nbsp;
>
> - Question 3: The reviewer notes that section 5 seems to be a draft without a comprehensive evaluation, and that some details are not clear, such as the prompt used to generate text.
>
>     &nbsp;
>
>    The main takeaway of section 5 is to show that bias propagates through training, such that a classifier trained on original data can easily distinguish generated data from LLMs trained on original data. We have made that clear in the revised paper, thanks for pointing it out.
>
>     &nbsp;
>
>     We also added section 6 in the revised paper, where we provided a comprehensive evaluation of bias propagation on several datasets, and showed that it can enable the estimation of the mixture proportions of the training domains of an LLM.
>
>     &nbsp;
>
>     Regarding the prompt, as explained in section 5, we prompt the LLMs with a single token, sampled from the distribution of tokens that appear as the first token in the sequences derived from original training data of the LLM. We prompt with only one single token, so that the LLM generates text unconditionally.
>
> &nbsp;
>
> We hope that those clarifications and the new results address the reviewer's concerns, and if yes, we would appreciate it if they would consider raising their score.

---

### Official Review · Reviewer_wbLi · 2024-10-31

**Soundness:** 2
**Presentation:** 1
**Contribution:** 2
**Rating:** 5
**Confidence:** 3

**Summary:**

This paper examines biases in popular pretraining datasets for large language models, demonstrating that transformer models can distinguish between texts from different datasets (like C4, RefinedWeb, and DolmaCC) with surprisingly high accuracy, despite these datasets being derived from CommonCrawl using similar filtering methods. Through user studies and rewriting experiments, the authors show these biases are subtle to humans but persistent through reformatting, and importantly, they propagate through training - models trained on these datasets inherit their distinctive characteristics. The work includes comprehensive ablation studies and extends similar dataset bias research from computer vision.

**Strengths:**

1. The paper shows how different filtering pipelines create distinct "fingerprints" in the data, even when using similar preprocessing steps.

2. The paper does comprehensive ablation studies examining key factors like model size, training data amount, and sequence length. These controlled experiments help isolate the important variables affecting classification accuracy. The validation approach using multiple methods (human studies, rewriting experiments, bias propagation tests) strengthens the findings by showing the robustness of the results across different experimental paradigms.

**Weaknesses:**

1. It doesn't deeply analyze what features enable this classification. A feature importance analysis (e.g., using attention weights or gradient-based attribution methods) could reveal which textual patterns or structures the classifier relies on, providing actionable insights for dataset creators.

2. The rewriting experiments use only GPT-4 for text modification. Testing with multiple different LLMs would strengthen the finding that biases persist through rewriting. Additionally, more controlled rewriting experiments (e.g., systematically modifying specific text features like sentence length, vocabulary complexity, or discourse markers) could better isolate which characteristics contribute to dataset fingerprints.

3. While the paper demonstrates dataset biases exist and propagate, it doesn't propose concrete methods to mitigate them.

**Questions:**

1. How do you ensure the classification accuracy on generated text isn't simply detecting general "AI-generated text" patterns rather than dataset-specific biases?

2. Have you tested if these biases persist through fine-tuning or RLHF? This seems crucial given current LLM development practices.

---

> ### Author Response · Authors · 2024-11-22
> **Response to reviewer wbLi**
>
> Many thanks for mentioning that we do comprehensive ablation studies, that our findings are strengthened by multiple methods, and that our results are robust across different experimental paradigms.
>
> &nbsp;
>
> - Response to weakness 1: Regarding ``It doesn't deeply analyze what features enable this classification'', we now did extensive further analysis in Sections 4.4, 4.5, and 4.6 on what features enable this classification. We find that formatting, vocabulary, and content distributions are all characteristics that are different between the datasets and that contribute to the distinguishability of the datasets.
>
> &nbsp;
>
> - Response to weakness 2: Regarding rewriting with models other than GPT4-mini; we looked at multiple models initially (GPT3.5, GPT4, and GPT4-mini), and tuned our prompt carefully to work well with GPT4-mini. We went through a lot of the text manually to see whether the rewrites are as intended. We do not think that this experiment will benefit significantly from using another model for rewriting, but we started the process of rewriting with another model, and will add this to the paper once it is done.
>
> &nbsp;
>
> - Response to weakness 3: The reviewer notes that we do not propose methods to mitigate the bias. The focus of our paper is not to mitigate bias, but to demonstrate that biases can be detected via classification experiments on text datasets, and persist in the models that are trained on those datasets.
>
>     &nbsp;
>
>     Having a bias has a negative connotation, thus mitigating seems natural, but in our context this is not implied. For instance the dataset FineWeb-Edu is biased towards educational content, and can therefore perform well on reasoning and knowledge benchmarks.
>
> &nbsp;
>
> - Response to question 1: The reviewer asks how we know that the classification accuracy on the generated text is due to the biases propagating from the original datasets, and not AI-generated text patterns. We know that from the experiment “Classifying generated data with a model trained to distinguish the original data” in section 5. In this experiment, the classifier is trained only on original data, yet it can classify the generated data well. Since the classifier has not been trained on any generated data, it can only utilize the learnt patterns from the original data to classify the generated data.
>
> &nbsp;
>
> - Response to question 2: Regarding whether we tested if these biases persist through fine-tuning: Thanks for the suggestion, in the meantime we tested this and found that to some extent the biases persist, please see the new Section 5.1.
>
> &nbsp;
>
> We hope that those clarifications and the new results address the reviewer's concerns, and if yes, we would appreciate it if they would consider raising their score.

---

> > ### Author Response · Authors · 2024-11-28
> > **Update on reviewer wbLi's second weakness**
> >
> > Regarding reviewer wbLi's suggestion to test rewriting with different LLMs other than GPT-4, we tested rewriting with Qwen2.5-14B-Instruct using the same exact 3 prompts as with GPT-4. The accuracies are as follows:
> > - prompt1: 81.98%
> > - prompt2: 77.83%
> > - prompt3: 69.78%
> >
> > The accuracies are similar to those obtained with GPT-4 (section 4.3 in the paper).
> >
> > This outcome strengthens the finding that biases persist through rewriting.

---

### Official Review · Reviewer_Dx2D · 2024-11-03

**Soundness:** 3
**Presentation:** 3
**Contribution:** 3
**Rating:** 6
**Confidence:** 3

**Summary:**

This work investigates the distinguishability of a range of popular open-source pretraining text datasets derived from CommonCrawl, including C4, RefinedWeb, DolmaCC, RedPajama-V2, FineWeb, DCLMBaseline, and others. The study presents interesting findings: 1) a classifier trained on these datasets achieves high accuracy on held-out test data, despite humans finding the task challenging; 2) this distinguishability extends to models pre-trained on each dataset—specifically, a classifier trained on the original text datasets performs well in distinguishing between models pre-trained on these datasets.

**Strengths:**

- The research question is interesting and impactful to the LLM research direction.
- The authors conduct extensive experiments, such as the impact of text rewrite,  and draw their findings in a rigorous way.

**Weaknesses:**

- It would be insightful to further investigate whether distinguishability propagates to models fine-tuned on the same downstream task. For instance, if models are pre-trained on different text datasets but fine-tuned on the same dataset, will their behaviors remain distinguishable?
- Considering that the construction of the pre-training datasets involves only data filtration, without any modification or augmentation, and that these datasets share similar sources, it seems counterintuitive that they are distinguishable at the level of individual segments. Could the authors provide further explanation on this?
- The study claims the existence of dataset bias by demonstrating corpus distinguishability. It would be beneficial to identify and describe more explicit dimensions of bias, as this would offer clearer implications and impact.

**Questions:**

Refer to weaknesses.

---

> ### Author Response · Authors · 2024-11-22
> **Response to reviewer Dx2D**
>
> Many thanks for mentioning that our research question is interesting and impactful in the LLM research, that we conduct extensive experiments, and that we ``draw findings in a rigorous way'' .
>
> &nbsp;
>
> Response to reviewer Dx2D’s concerns:
>
> - Thanks for the suggestion to add an experiment on whether  bias propagates through finetuned models. We added this experiment in Section 5.1.
>
> &nbsp;
>
> - Regarding further explanation on what makes the datasets distinguishable and and what are explicit  biases or differences, we added the new sections 4.4, 4.5, and 4.6 to the revised paper, where we investigate formatting, word distributions, and topics as sources of bias/difference and find that each of those are different, but do not account alone for the distinguishability.
>
> &nbsp;
>
> We hope that those clarifications and the new results address the reviewer's concerns, and if yes, we would appreciate it if they would consider raising their score.

---

> ### Author Response · Authors · 2024-12-02
>
> We would like to thank reviewer Dx2D once again for their valuable feedback and suggestions. As we approach the end of the discussion period, we hope our responses and new results have addressed the reviewer's concerns. If so, we would kindly ask the reviewer to consider reflecting this in their score. If there are any remaining points of clarification or further questions, we would be more than happy to provide additional explanations.

---

### Official Review · Reviewer_iJhV · 2024-11-04

**Soundness:** 3
**Presentation:** 3
**Contribution:** 3
**Rating:** 5
**Confidence:** 3

**Summary:**

This paper investigates the biases present in LLM pretraining datasets and examines how these biases persist and propagate through training. The study calms different datasets possess unique biases or fingerprints identifiable by models, even when preprocessed similarly or rewritten. Shows that classifiers can distinguish dataset origin with high accuracy, and biases can carry over.

**Strengths:**

1. The study provides a detailed look at biases in seven widely used LLM pretraining datasets. Revealed biases persist even when text is rephrased by other LLMs.

2. By showing that dataset biases are measurable, persistent, and propagate into LLM-generated outputs. It suggests that even datasets created with strict filtering and deduplication standards still exhibit biases, emphasizing the need for new methods to mitigate these issues.

**Weaknesses:**

1. My main concern is the study use of prompt-based rephrasing to test bias persistence introduces potential confounding effects, as prompts may inadvertently impose their own linguistic patterns or styles. This prompt influence could create artifacts that the classifier detects, rather than the underlying biases in the original datasets.

2. The study sticks mostly to a 160M model, barely looking into how bigger models, like the billion-parameter ones used in real applications, might handle and spread dataset biases.  Without testing scalability, it’s unclear if the study’s conclusions hold for bigger, more powerful models where bias effects could still remain or reduce.

**Questions:**

see above

---

> ### Author Response · Authors · 2024-11-22
> **Response to reviewer iJhV**
>
> Thanks for recognizing that our study provides a detailed look at 7 datasets and shows that dataset biases are measurable, persistent, and propagate through LLM outputs.
>
> &nbsp;
>
> Response to reviewer iJhV’s concerns:
>
> - The reviewer’s main concern is that the prompt-based rephrasing might introduce bias in the data which the classifiers detect rather than the bias in the original data. The main results in Table 1 are all on original data, without any rephrasing, thus the classifiers detect underlying biases in the original datasets.
>
>     &nbsp;
>
>     Regarding rephrasing, we agree with the reviewer that the rephrasing model (GPT-4o-mini) might induce biases in the output. However   since we use the same model and prompt for rephrasing, if the bias were very strong, it would make the datasets very difficult to distinguish after rewriting. What we see, however, is that the data remains distinguishable even after rewriting.
>
>     &nbsp;
>
>     The rephrasing experiment is to investigate what makes the datasets different. We also added several new experiments in Section 4.4, 4.5, 4.6 to further understand which aspects make them distinguishable, for example in section 4.4 of the revised paper, we removed formatting while keeping the wording exactly the same (no rephrasing). This helps isolate the effect of format without any LLM induced bias.
>
>     &nbsp;
>
> - Regarding that we use a 160M model, without looking into larger models with billions of parameters used in real applications. Please note that we use the 160M model only as  a classifier, but we do study billion parameter models when rephrasing and generating data (Sections 4.3 and 5).
>
>     &nbsp;
>
>     For classification there is little to no benefit when using larger models, as our ablation study in Figure 1 shows. Specifically, the classification accuracy for model sizes ranging from 25M to 410M only differs by 0.56%, as discussed in section 4.1.
>
>     &nbsp;
>
> We hope that those clarifications and the new results address the reviewer's concerns, and if yes, we would appreciate it if they would consider raising their score.

---

> ### Author Response · Authors · 2024-12-02
>
> We would like to thank reviewer iJhV once again for their valuable feedback and suggestions. As we approach the end of the discussion period, we hope our responses and new results have addressed the reviewer's concerns. If so, we would kindly ask the reviewer to consider reflecting this in their score. If there are any remaining points of clarification or further questions, we would be more than happy to provide additional explanations.

---

### Author Response · Authors · 2024-11-22
**Common comments to all reviewers and AC**

We would like to thank all reviewers for their valuable feedback that has helped refine the paper. Here are the changes we made to the paper following the reviewer’s suggestions:

- Reviewers Dx2D, wbLi, and UcRC suggested that more insights into the features that enable classification between the datasets would be helpful. We added experiments on removing formatting, classifying based on frequency of words, and dataset content categorization, which together suggest that formatting, vocabulary, and content distributions are all characteristics that lead to differences between the datasets.   We also provided concrete examples for particular patterns within some datasets. Please refer to the new sections 4.4, 4.5, 4.6 in the revised paper for a detailed description.

- Reviewers Dx2D and wbLi suggested an experiment with instruction finetuning to investigate if bias still propagates through finetuned models. We added an experiment for finetuning, which shows that bias still persists even in instruction finetuned models, albeit less than in the original pretrained model. Please refer to section 5.1 in the revised paper for more details.

- Reviewer UcRC suggested to do  a more comprehensive evaluation of bias propagation on other datasets. We added experiments on more datasets, and showed that bias propagation can enable the estimation of mixture proportions of the training domains of an LLM. Please refer to section 6 in the revised paper for details.

- Reviewer UcRC requested 2-way classification experiments for all possible 21 binary combinations between the seven datasets. We added the classification accuracies in appendix B in the revised paper.

Other minor changes to the paper:

- For the experiment “Classifying generated data with a model trained to distinguish the original data” in section 5, we previously used the OLMo-7B model to generate data, which is trained on all domains of the Dolma dataset (the exact ratio from each domain is not known). The classifier, however, was only trained on the DolmaCC domain. This experiment had an accuracy drop of about 9% from original to generated data, which we previously attributed to the mismatch between generated and original data. In the revised paper, we replaced OLMo-7B with Falcon-7B, which is trained on RefinedWeb (exists as a single domain). The accuracy drop in this case is less than 1%, showing that the previous accuracy drop was majorly due to the inconsistency of the training data between the classifier and the LLM rather than the mismatch between the original and generated data. This outcome strengthens the finding that bias propagates through training.

- We increased the training tokens and test sequences of the rewritten and generated data experiments to 160M training tokens and 8192 test sequences, for consistency with the other experiments on the original data throughout the paper.

- We added an ablation study in Appendix C with BERT as a classifier, which showed to perform similarly to the autoregressive transformer.

We also respond to each reviewer individually below. We hope we were able to address the concerns from all reviewers, and are happy to clarify further. We hope the reviewers reassess their evaluations after reading our responses!

---

### Author Response · Authors · 2024-11-28
**Follow-Up on revised paper and reviewer feedback**

We sincerely thank reviewers wbLi and UcRC for raising their scores after reviewing our responses and the revised paper. We would also be grateful if reviewers iJhV and Dx2D might consider raising their scores if our responses and new results have adequately addressed their concerns. If there is anything that remains unclear, we would be happy to provide further clarification.

---

### Meta-Review · Area_Chair_wZ7T · 2024-12-20

**Metareview:**

This paper discusses the bias present in large-scale text datasets used for pretraining LLMs. The analysis shows that it is possible to distinguish datasets with a simple classifier with relatively high accuracy. Moreover, the bias propagates to generated content and is not easily removed by AI-based paraphrasing. While the topic is interesting, the execution of this work could be improved. Better quantitative and qualitative analysis of how does the bias actually look like should be provided. Moreover, the paper does not provide a clear answer to the "so what?" question. In its current form, this work would be a better fit for a specialized workshop about training data. I recomment this paper for rejection.

**Additional Comments On Reviewer Discussion:**

The reviewers raised their scores after rebuttal, with a 3->5 and a 5->6. This changed the ratings from 3556 to 5566. Despite this increase, I think this paper does not meet the bar of ICLR 2025.

---

### Decision · Program_Chairs · 2025-01-22

Reject